

# EPiC-GAN: Equivariant point cloud generation for particle jets

Erik Buhmann[1⋆], Gregor Kasieczka[1,2] and Jesse Thaler[3,4]

1 Institut für Experimentalphysik, Universität Hamburg, Germany
2 Center for Data and Computing in Natural Sciences (CDCS), Hamburg, Germany
3 Center for Theoretical Physics, Massachusetts Institute of Technology,
Cambridge, Massachusetts, United States
4 The NSF AI Institute for Artificial Intelligence and Fundamental Interactions

⋆ erik.buhmann@uni-hamburg.de

## Abstract

With the vast data-collecting capabilities of current and future high-energy collider experiments, there is an increasing demand for computationally efficient simulations. Generative machine learning models enable fast event generation, yet so far these approaches are largely constrained to fixed data structures and rigid detector geometries. In this paper, we introduce EPiC-GAN — equivariant point cloud generative adversarial network — which can produce point clouds of variable multiplicity. This flexible framework is based on deep sets and is well suited for simulating sprays of particles called jets. The generator and discriminator utilize multiple EPiC layers with an interpretable global latent vector. Crucially, the EPiC layers do not rely on pairwise information sharing between particles, which leads to a significant speed-up over graph- and transformer-based approaches with more complex relation diagrams. We demonstrate that EPiC-GAN scales well to large particle multiplicities and achieves high generation fidelity on benchmark jet generation tasks.

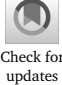

# 1   Introduction

The simulation of particle interactions is a crucial part of fundamental physics research. Such simulations are essential for comparing theoretical predictions to experimental results. There is a growing interest in augmenting or replacing these simulations with generative machine learning models; see Ref. [1] for a recent review. An important motivation is the ever-increasing need for large-scale simulation samples. With the high-luminosity upgrade of the Large Hadron Collider (HL-LHC) and anticipated future collider projects, there will be a significant increase in the volume of experimental data which will need to be matched by simulations.

Generative machine learning models can be used to speed up simulations by leveraging advances in efficient GPU computations and thereby mitigating the need for computationally expensive Monte Carlo sampling. Such an upfront central generative model training can minimize local computing resources needed to reproduce a large number of events. For these applications, the generative models need to learn the underlying data distributions with help from the inductive bias of the model architecture. In this way, it is possible to amplify the training statistics [2,3] as well as utilize generative modeling as a powerful data augmentation technique [4].

Some very advanced generative modeling efforts for high-dimensional data in high energy physics (HEP) are currently undertaken for calorimeter shower simulations. These fast simulations show ever-increasing fidelity for low- and high-granular calorimetry. Various approaches have been used for these generative models, including generative adversarial networks (GANs) [5–10], autoencoders [11–14], normalizing flows [15, 16], and score-based models [17]. The ATLAS experiment at the LHC is already using generative models in their analysis pipeline [10]. Here, we introduce a novel kind of GAN architecture which could be applied to various applications. Of course, when applied to a specific physics analysis, the uncertainties derived from a generative model need to be critically examined depending on the use case [18].

Previous work on generative models for HEP has focused primarily on simulating rigid detector geometries. The corresponding models are largely constrained to fixed data structures, such as images or fixed-length lists. For many experimental setups, including for collider experiments, the geometric position of sensor data is heterogeneous (unlike an image) and the number of sensor outputs is variable. In this context, the data associated with the sensor array is a *set* of geometric positions and sensor readouts, such as light-yields or energies. The natural representation of this kind of experimental data is an unordered, variable-size point cloud, which can be viewed as a graph of nodes without edges. Such data representations warrant permutation equivariant generative models, like set- and graph-based networks, that do not rely on a human-biased ordering and naturally work with inputs of variable cardinality. While generative models using fully-connected network architectures work for certain applications [19, 20], they are less generalizable due to the model constraints.

Recent work on permutation equivariant generative models for point cloud data has focused on graph- and transformer-based architectures [21–27]. The current state-of-the-art for point cloud data generation in HEP is the message-passing GAN (MP-GAN), which has been used to generate particle jets [24]. This model utilizes multiple fully-connected message-passing neural network (MPNN) blocks [28] in both the discriminator and the generator. These blocks facilitate pairwise message passing between all points and are therefore capable of computing complex functions on the entire graph. A key drawback of the MP-GAN approach, though, is that the number of passed messages scales quadratically with the number of points, as do the computational requirements. This is true for any graph-network or transformer-based generative model that utilizes pairwise edge attributes.

In this paper, we introduce a generative model for point clouds that uses global attributes and point attributes, without computing any pairwise edge features. Our equivariant point cloud GAN (EPiC-GAN) is based on the framework of deep sets [29], known in the HEP community as particle flow networks (PFNs) [30], whose computational cost scales linearly in the number of points. We find that EPiC-GAN provides fidelity of generated distributions on par with MP-GAN, yet offers a significant speed-up in generation time and much better scaling to large point cloud multiplicities. Additionally, the global attributes associated with each EPiC-GAN layer provide an interpretable latent space that can be correlated to known physical observables.

As a proof-of-principle case study, we apply EPiC-GAN to generate jets trained on the JetNet benchmark dataset [24]. JetNet is a specifically designed toy dataset for the generation of hadronized jets used to compare point cloud generative models. In comparison to earlier permutation equivariant set-based generative models [31–33], the EPiC-GAN utilizes a continuously updated global attribute vector that allows for inter-point communication without the computational overhead of a full graph model. A model relying on such global attributes works well for modeling particle jets, which are defined by a number of per-jet (i.e. global) physical observables such as mass and transverse momentum. The fidelity reached on such complex physical distributions suggests that this model could also perform well for tasks like fast calorimeter simulation. We further envision that our model could be applied to generate in-situ background events for anomaly detection methods such as CATHODE [34]. Additionally, the development of fast and lightweight generative models can support analysis by making Monte Carlo tuning or nuisance parameter variation computationally efficient.

The remainder of this paper is organized as follows. In Sec. 2, we introduce the EPiC-GAN architecture and associated loss functions. We present a case study using EPiC-GAN for generating jets at the LHC in Sec. 3 and draw our conclusions in Sec. 4.

## 2 Equivariant point cloud GAN

In this section, we introduce equivariant point cloud (EPiC) layers, which are the foundation for our generative model. By stacking multiple EPiC layers, we build our generator and discriminator architectures. To the best of our knowledge, these architectures are a novel contribution to the generative modeling literature, not just in HEP. We implement EPiC-GAN in Pytorch [35] and the code is available on GitHub.[1]

### 2.1 EPiC layers

Following the notation in Ref. [36], we define a 2-tuple point cloud $C = (\boldsymbol{g}, P)$ as a graph without edges. The global attributes of the point cloud are represented by $\boldsymbol{g}$. The set of points

---

[1]https://github.com/uhh-pd-ml/EPiC-GAN.

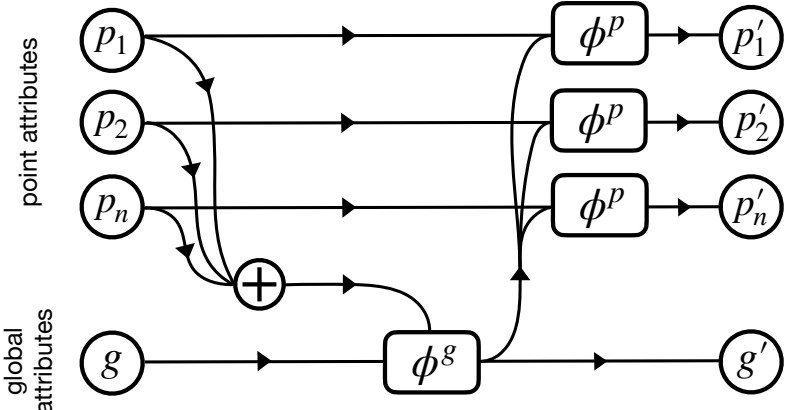

Figure 1: Equivariant Point Cloud (EPiC) layer structure. The global function $\phi^g$ and point function $\phi^p$ are learned by neural networks. The $\oplus$ symbol indicates the aggregation function $\rho^{p \to g}$ with both element-wise summation and average pooling.

are represented by $P = \boldsymbol{p}_{i=1:N}$, where $\boldsymbol{p}_i$ are the attributes of point $i$ and $N$ is the set cardinality (in jet physics terms: particle multiplicity).

The EPiC layer transforms both the global attributes $\boldsymbol{g} \to \boldsymbol{g}'$ and the set of points $P \to P'$ according to the diagram in Fig. 1. This transformation is permutation equivariant by construction and involves two consecutive computation:

$$\boldsymbol{g}' = \phi^g\left(\boldsymbol{g}, \rho^{p \to g}(P)\right), \tag{1}$$

$$\boldsymbol{p}_i' = \phi^p(\boldsymbol{g}', \boldsymbol{p}_i). \tag{2}$$

The global functions $\phi^g$ and point function $\phi^p$ are learned by neural networks, which for concreteness we take to be a 2-layer Multilayer Perceptrons (MLPs) with LeakyReLU activation functions. The aggregation function $\rho^{p \to g}$, which involves a concatenation of both element-wise summation and average pooling, maps the point information into a common global feature.

As with all set- and graph-based networks such as Ref. [28, 29, 37], the EPiC layer can be seen as a specific case of the Graph Network Block in Ref. [36], with the following changes:

- No edge features are used. As discussed in the introduction, this ensures that the computational cost of an EPiC layer scales linearly with the number of points.

- The global attributes transformation $\phi^g$ is performed before the point attributes transformation $\phi^p$. This ensures that even with one EPiC layer, the point transformation has access to global information.

- The aggregation function $\rho^{p \to g}$ uses both mean and sum pooling. We found empirically that both pooling operations were necessary for good performance on variable-sized point clouds. Summation as an injective aggregation operator preserves set cardinality. For large cardinality variance, though, mean aggregation supports faster model convergence since the scale of the output is independent of cardinality.

To the best of our knowledge, this is the first time a graph network block has been proposed with this specific structure.

By limiting the number of global attributes (the dimensionality $\dim(\boldsymbol{g})$), the amount of communication between points $\boldsymbol{p}_i$ is bottlenecked. Therefore, the number of stacked EPiC-layers $L$ and $\dim(\boldsymbol{g})$ are two hyperparameters that can be used to optimize a model for the

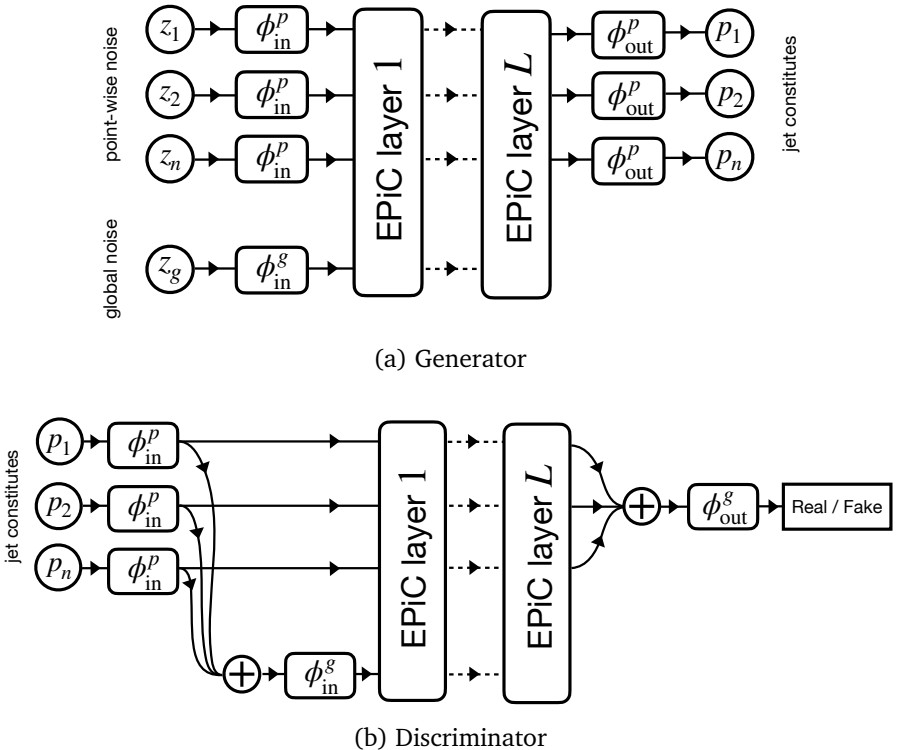

(a) Generator

(b) Discriminator

Figure 2: Architecture implementation of the EPiC GAN. Both the (a) generator and (b) discriminator consist of multiple EPiC layers from Fig. 1 as well as (shared) neural networks for input/output dimensionality expansion/reduction. The ⊕ symbol represents the aggregation function $\rho^{p \to g}$ with both element-wise summation and average pooling. Though not shown, there are additional residual connections between EPiC layers described in the text.

minimum amount of inter-point communication necessary to perform a given task. As discussed in Sec. 3.6, this global bottlenecking also facilitates interpretation of the network in terms of learned global features.

## 2.2 GAN architecture

Like all GANs, EPiC GAN feature a generator and discriminator, whose specific architectures are shown in Fig. 2. Both networks consist of multiple consecutive EPiC layers that facilitate point cloud transformations based on a global attribute vector. In the discriminator, two additional aggregation functions $\rho^{p \to g}$ are inserted to perform mean and sum pooling. This aggregation is used before the EPiC layers to create a global attribute vector from the jet constituents, and it is used after the EPiC layers to create a permutation invariant discriminator output.

To further improve the performance, the generator uses input ($\phi_{\text{in}}^{p}$) and output ($\phi_{\text{out}}^{p}$) blocks for dimensionality expansion/reduction of the point-wise information. These blocks are learned by a shared 1-layer MLP with LeakyReLU activation function, which allow the EPiC layers to act on a larger hidden dimensionality. Similarly, the block $\phi_{\text{in}}^{g}$ transforms the global input noise and is implemented with a 2-layer MLP. The discriminator has the same input/output block structure, except there is only one shared 2-layer MLP $\phi_{\text{in}}^{p}$ and two global 2-layer MLPs $\phi_{\text{in}}^{g}$ and $\phi_{\text{out}}^{g}$. The latter yields a single output to discriminate between real and fake jets. Though not shown in Fig. 2, we add a residual connection [38] to every EPiC layer as well as a residual connection between the input of the first EPiC layer to the output of every EPiC layer.

During the training, the training set is divided into batches of jets with equal particle multiplicity and a set maximum batch size (128 for all experiments). This avoids needing to zero-pad jets and it ensures that the discriminator always compares jets with the same particle multiplicity. As input to the generator, the global attribute vector is sampled from a normal distribution with $\mathcal{N}(0,1)$ and a dimensionality $\dim(\boldsymbol{g})$. The particle multiplicity is specified at the noise sampling step when generating a batch of jets, with the point-wise noise sampled from $\mathcal{N}(0,1)$ with the same dimensionality as the jet constituents (3 for the case study below). During the evaluation, the particle multiplicity is sampled from a Kernel Density Estimation (KDE) of the particle multiplicity distribution of the training set.

The GAN training objective follows the Least Squares GAN (LSGAN) approach using the least squares loss function [39]. The loss functions $L(D)$ and $L(G)$ of the discriminator $D$ and the generator $G$ respectively are given by:

$$\min_{D} L(D) = \frac{1}{2}\mathbb{E}_{\boldsymbol{x}\sim p_{\text{data}}(\boldsymbol{x})}\left[(D(\boldsymbol{x})-1)^2\right] + \frac{1}{2}\mathbb{E}_{\boldsymbol{z}\sim p_{z}(\boldsymbol{z})}\left[(D(G(\boldsymbol{z})))^2\right], \tag{3}$$

$$\min_{G} L(G) = \frac{1}{2}\mathbb{E}_{\boldsymbol{z}\sim p_{z}(\boldsymbol{z})}\left[(D(G(\boldsymbol{z}))-1)^2\right]. \tag{4}$$

We compared this approach to the regular GAN objective using the binary cross-entropy loss and found slightly improved training stability with the LSGAN objective as it avoids vanishing gradients. Additionally, we apply weight normalization [40] to all hidden layers of both the generator and discriminator to improve the GAN stability.

# 3 Case study in jet physics

We now apply EPiC-GAN to a specific task in particle physics: the generation of particle jets for different jet types with variable particle multiplicities. Generating particle jets from first principles is a well-understood process and many particle- and (sub)jet-level observables are straightforward to calculate, resulting in a sound test space to benchmark generative models.

Our case study is based on the JetNet datasets [41, 42], which were specifically designed to compare generative models for equivariant point clouds with variable cardinality in particle physics. After outlining the specific EPiC-GAN architecture and training procedure, we present our results on the JetNet30 and JetNet150 datasets. We then highlight the fast and scalable nature of EPiC-GAN and discuss the interpretability of its global latent space.

## 3.1 JetNet datasets

The JetNet30 [41] and JetNet150 [42] benchmark datasets were generated with the PYTHIA 8.212 [43] parton shower event generator. Hadronized jets from proton-proton collisions at 13 TeV are clustered with the anti-$k_{\text{T}}$ algorithm [44] with a jet radius of $R = 0.8$. The particle collider coordinates are normalized and centered resulting in three particle features: the relative transverse momentum $p_{\text{T}}^{\text{rel}} = p_{\text{T}}^{\text{particle}}/p_{\text{T}}^{\text{jet}}$, the relative pseudorapidity $\eta^{\text{rel}} = \eta^{\text{particle}} - \eta^{\text{jet}}$, and the relative azimuthal angle $\phi^{\text{rel}} = \phi^{\text{particle}} - \phi^{\text{jet}}$. The particles are then $p_{\text{T}}$ sorted and the leading 30 particles are used. A detailed description of the JetNet30 dataset as well as an implementation of multiple point cloud generative networks can be found in Ref. [24]. The JetNet150 dataset follows the same design, except that the leading 150 particles are used.

For our study, we separately consider three JetNet datasets with different initiating partons: gluon jets, light quark jets, and top jets. All datasets consist of about 170,000 events. We split the dataset as 70%/15%/15% into a training set of about 120,000 events and a validation and test set each of about 25,000 events. The validation set is used to choose the best epoch while

Table 1: Hyperparameter choices for the EPiC-GAN used in the JetNet case study.

| Hyperparameter | Value |
|---|---|
| $L_{\text{generator}}$ EPiC layers | 6 |
| $L_{\text{discriminator}}$ EPiC layers | 3 |
| $\dim(\boldsymbol{g})$ global dimensionality | 10 |
| $\dim(\boldsymbol{p}_i)$ point dimensionality | 3 |
| Hidden dimensionality | 128 |
| Activation function | LeakyReLU(0.01) |
| Adam [45] learning rate | $10^{-4}$ |
| Batch size | 128 |
| Training epochs | 2000 |
| Generator weights | $\sim 425{,}000$ |
| Discriminator weights | $\sim 313{,}000$ |

the test set is used in Secs. 3.3 and 3.4 to report the evaluation scores and shown distributions.[2] Because of the artificial restriction to 30 and to 150 particles, the jets are no longer centered, so we added an extra pre-processing step that re-centers the jets based on kept particle subset.

## 3.2 Architecture & training procedure

For each of the six datasets (gluon, light quark, and top; each with a maximum of 30/150 particles, respectively) we train a separate EPiC-GAN for 2000 epochs. The EPiC-GAN generators are implemented with $L_{\text{generator}} = 6$ EPiC layers, and the discriminators use $L_{\text{discriminator}} = 3$ EPiC layers. The dimensionality of the input noise was set to three noise variables per point ($\dim(\boldsymbol{p}_i) = 3$), to match the three-dimensional particle features space of ($p_{\text{T}}^{\text{rel}}, \eta^{\text{rel}}, \phi^{\text{rel}}$). We standardized the features of the training set to follow a normal distribution with $\mathcal{N}(0, 5^2)$, i.e. 5 times wider than a unit Gaussian, which we found to improve the discriminator performance. The generated jets are post-processed by centering them as well as setting all particle $p_{\text{T}}$ values below the minimum $p_{\text{T}}$ value of the training set to that value. Further implementation details and hyperparameters can be found in Table 1. Due to computational costs, we did not perform a large-scale hyperparameter scan. In principle, the discriminator could also be implemented with any other permutation-equivariant graph- or set-based architecture. However, replacing the EPiC discriminator with a simple Particle Flow Network [30] – equivalent to $L_{\text{discriminator}} = 0$ EPiC layers – led to worse results.

Similar to Ref. [24], the best epoch for each model is chosen based on the mean of the Wasserstein-1 distance of the relative jet mass distribution $W_1^M$ between the validation set and 10 different generated jet sets of the same size ($\approx 25{,}000$ events). We chose the jet mass because a well-described mass distribution correlates well with a good modeling of other jet features, for example, Energy Flow Polynomials (EFPs) [46]. Because of the adversarial training, GANs are inherently difficult to train, and we observed slight deviations between the convergence behavior and generative fidelity when training the EPiC-GAN multiple times. We opted to train the EPiC-GAN three times for each dataset and show here the models with the lowest $W_1^{\text{mass}}$ score in the validation set.

Note that many alternative methods to choose the best epoch could be possible, such combining comparison scores of different 1-dimensional histograms [12] or using a single method (i.e. MMD or Wasserstein-$p$ distance) to evaluate a multidimensional set of observables [26].

---

[2]Ref. [24] used a 70%/30% training/test set split, and the test set was used to determine the best epoch.

Table 2: Evaluation scores for the JetNet30 dataset. The truth values are a comparison between the test and training set, which reflect the size of statistical fluctuations. The MP-GAN scores were calculated with the trained models from Ref. [24] using the same statistics as the EPiC-GAN. Lower is better for all scores.

| Jet class | Model | $W_1^{\mathrm{M}}$ ($\times 10^{-3}$) | $W_1^{\mathrm{P}}$ ($\times 10^{-3}$) | $W_1^{\mathrm{EFP}}$ ($\times 10^{-5}$) | FPND |
|---|---|---|---|---|---|
| Gluon | Truth | $0.3 \pm 0.1$ | $0.3 \pm 0.1$ | $0.3 \pm 0.3$ | $0.07 \pm 0.01$ |
| | MP-GAN | $0.5 \pm 0.1$ | $\mathbf{1.3 \pm 0.1}$ | $0.6 \pm 0.3$ | $\mathbf{0.13 \pm 0.02}$ |
| | EPiC-GAN | $\mathbf{0.3 \pm 0.1}$ | $1.6 \pm 0.2$ | $\mathbf{0.4 \pm 0.2}$ | $1.01 \pm 0.07$ |
| Light quark | Truth | $0.3 \pm 0.1$ | $0.3 \pm 0.1$ | $0.3 \pm 0.3$ | $0.02 \pm 0.01$ |
| | MP-GAN | $\mathbf{0.5 \pm 0.1}$ | $4.9 \pm 0.3$ | $\mathbf{0.7 \pm 0.4}$ | $\mathbf{0.36 \pm 0.02}$ |
| | EPiC-GAN | $0.5 \pm 0.1$ | $\mathbf{4.0 \pm 0.4}$ | $0.8 \pm 0.4$ | $0.43 \pm 0.03$ |
| Top | Truth | $0.2 \pm 0.1$ | $0.3 \pm 0.1$ | $0.6 \pm 0.5$ | $0.02 \pm 0.01$ |
| | MP-GAN | $\mathbf{0.5 \pm 0.1}$ | $2.4 \pm 0.2$ | $\mathbf{1.0 \pm 0.7}$ | $0.35 \pm 0.04$ |
| | EPiC-GAN | $0.5 \pm 0.1$ | $\mathbf{2.1 \pm 0.1}$ | $1.7 \pm 0.3$ | $\mathbf{0.31 \pm 0.03}$ |

Either way, the question arises of which physically motivated or machine learned set of observables should be used for evaluation. The question of choosing the "best" epoch in a GAN is tightly correlated to the question of choosing the "best" generative model for a given task. We believe further research is needed on evaluation metrics for generative models in the direction of work such as Ref. [26] to advance the field of generative machine learning in HEP. Additionally, a sufficient amount statistics and a sound error estimation needs to be agreed upon. Here, we sidestep this discussion by using a similar method as Ref. [24] and are therefore able to compare our results with the MP-GAN approach.

## 3.3 JetNet30 results

We now compare the EPiC-GAN results on the JetNet30 dataset to the current state-of-the-art generative model for equivariant and unconditional generation, the MP-GAN [24]. The recently published generative adversarial particle transformer (GAPT) [26] is not included in the comparison, since it under-performs MP-GAN. We also do not compare to the recent JetFlow network [20], since it does not allow for equivariant generation and requires explicit conditioning on the jet mass and particle multiplicity, reducing its general applicability. For our comparison, we used the trained weights of the MP-GAN models published alongside Ref. [24]. Additionally, we applied the centering mentioned in Sec. 3.1 to the MP-GAN generated events, though this did not influence any of the evaluation scores or plots.

### 3.3.1 Evaluation scores

We start by giving an overview of the EPiC-GAN performance, by using the multiple evaluation scores introduced in Ref. [24]. In Table 2, we compare the EPiC-GAN fidelity to the JetNet truth and the MP-GAN.

Three evaluation scores are calculated with the Wasserstein-1 distances $W_1$ between different jet- and particle-level observables. The $W_1^{\mathrm{M}}$ score is the distance between the relative jet mass $m_{\mathrm{jet}}^{\mathrm{rel}}$ distributions of the test set and a generated dataset of the same size. The $W_1^{\mathrm{EFP}}$ score is the average distance between the distributions of five Energy Flow Polynomials (EFPs) (the loop-less multi-graphs with 4 nodes and 4 edges). The $W_1^{\mathrm{P}}$ score corresponds to the average distance between the particle feature distributions $p_{\mathrm{T}}^{\mathrm{rel}}$, $\eta^{\mathrm{rel}}$, and $\phi^{\mathrm{rel}}$. Additionally, the Fréchet

ParticleNet Distance (FPND), inspired by the Fréchet Inception Distance (FID), is calculated with its pre-trained implementation in the JETNET Python package as introduced in Ref. [24]. As of this writing, the FPND was only available for the JetNet30 dataset, and not JetNet150.

In comparison to the scores introduced in Ref. [24], we make slight changes to the evaluation procedure to increase the statistical significance of our reported scores and errors. For the "truth" scores, we compare the test set to the training set. We report the mean and the standard deviation for each score when calculated based on the test set and four different subsets of the training set with the same size as the test set ($\approx 25,000$ events). Similarly, the FPND score represents the mean and standard deviation of calculating the FPND based on these four training subsets. For the MP-GAN and the EPiC-GAN scores, we report the mean and standard deviation from comparing the test set to 10 different sets of generated events. The FPND was calculated with these 10 generated sets, too.[3]

The comparison in Table 2 shows that for almost all the Wasserstein distances, the EPiC-GAN performs equally good as the MP-GAN. For most scores, the EPiC-GAN and the MP-GAN lie within the margin of error. The exceptions are the light quark $W_1^{\mathrm{P}}$, which is slightly better for the EPiC-GAN, and the top $W_1^{\mathrm{EFP}}$, which is slightly better for the MP-GAN. In the FPND score, the MP-GAN out-performs the EPiC-GAN in the gluon and light quark jet classes, while the EPiC-GAN is better for the top quark dataset. This score and its scaling is however hard to interpret, since it is not clear which physical features the ParticleNet learns to highlight. Additionally, the pre-processing outline in Sec. 3.1 might have slight influence on the FPND score. Note that other tested GAN architectures in Ref. [24] performed significantly worse in the FPND score.

Compared to the truth scores, the GANs perform worse on almost all scores, except for the gluon and light quark EFP-based scores and the EPiC-GAN mass scores, where they perform within the margin of error from the truth. Depending on the specific application, this level of accuracy may or may not be sufficient. There is room for improvement and we hope future developments in generative modeling will yield even better models.

While the EPiC-GAN performs very similarly in generation fidelity as the MP-GAN, its advantage lies in the scaling behavior to large cardinalities, as emphasized in Sec. 3.5 below. The EPiC-layer structure scales $\approx \mathcal{O}(N)$ while the MP-GAN as a fully-connected graph network scales with $\approx \mathcal{O}(N^2)$. For this reason, we are able to show results for the larger JetNet150 dataset in Sec. 3.4.

### 3.3.2 Gluon dataset

We now take a closer look at different particle- and jet-level distributions for the three jet types, starting with gluon jets. In Fig. 3, we compare the JetNet30 gluon test set to the same number of GAN generated events. The EPiC-GAN is trained as detailed above. For the MP-GAN, we use the published weights from Ref. [24] to generate the same number of jets as in the test set. For all datasets, we compare a total of nine distributions: the relative particle features (relative transverse momentum $p_{\mathrm{T}}^{\mathrm{rel}}$, relative pseudorapidity $\eta^{\mathrm{rel}}$, relative azimuthal angle $\phi^{\mathrm{rel}}$); the 1st, 5th, and 20th leading particle $p_{\mathrm{T}}^{\mathrm{rel}}$; and three jet observables (particle multiplicity $N$, relative jet mass $m_{\mathrm{jet}}^{\mathrm{rel}}$, relative transverse momentum of the jet $p_{\mathrm{T,jet}}^{\mathrm{rel}}$).

Overall, both the EPiC-GAN and the MP-GAN generated events agree well with the Jet-Net30 gluon test set. Comparing the particle-level distributions, both GANs model the $p_{\mathrm{T}}^{\mathrm{rel}}$ and $\eta^{\mathrm{rel}}$ distributions equally well, while the EPiC-GAN represents the $\phi^{\mathrm{rel}}$ spectrum slightly better. Both models struggle to represent the outliers in the $\eta^{\mathrm{rel}}$ distribution.

---

[3]The MP-GAN scores in Ref. [24] were calculated with different statistics and therefore deviate slightly from our calculated scores.

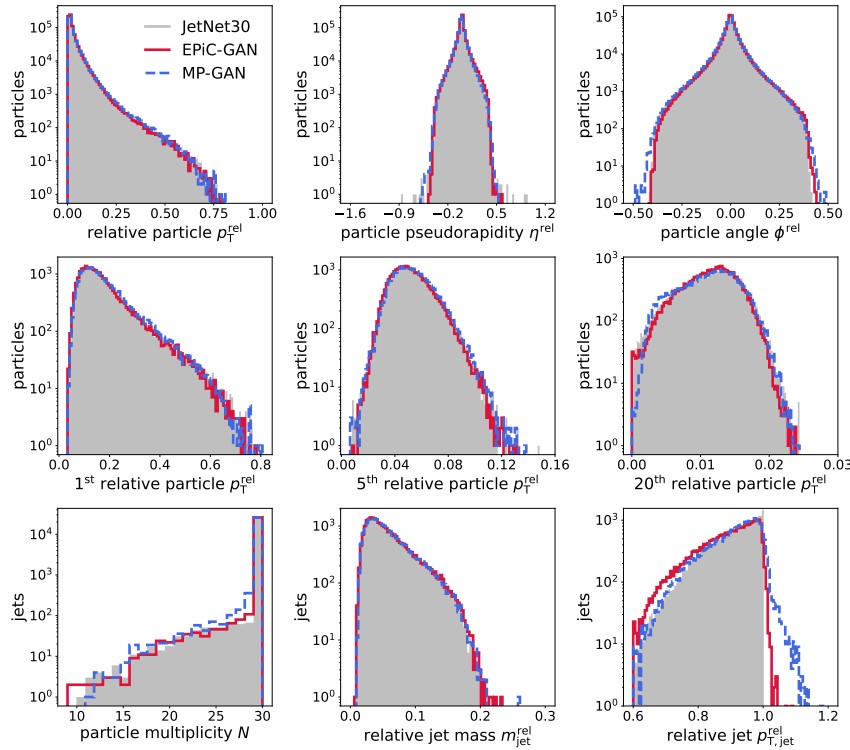

Figure 3: JetNet30 gluon dataset. Comparison between the test set, MP-GAN generated and EPiC-GAN generated sets for various particle- and jet-level observables.

For the 1$^{\text{st}}$ and 5$^{\text{th}}$ $p_{\text{T}}^{\text{rel}}$, both models are hardly distinguishable from the truth, and for the 20$^{\text{th}}$ $p_{\text{T}}^{\text{rel}}$, both models fall slightly short of modeling the low $p_{\text{T}}^{\text{rel}}$ part of the distribution.

The particle multiplicity distribution is handled well by both the MP-GAN and the KDE-based sampling method of the EPiC-GAN. Both GANs excel in representing the $m_{\text{jet}}^{\text{rel}}$ distribution as this was the basis for choosing each of the epochs. The $p_{\text{T,jet}}^{\text{rel}}$ spectrum is cut-off at exactly $p_{\text{T,jet}}^{\text{rel}} = 1$ due to the normalization applied first and the particle multiplicity cut second, as outlined in Sec. 3.1. Such a sharp cut-off is difficult for generative models to learn, therefore a tail towards higher $p_{\text{T,jet}}^{\text{rel}}$ for both GANs is to be expected, yet handled better by the EPiC-GAN. The low $p_{\text{T,jet}}^{\text{rel}}$ part of the spectrum is represented better by the MP-GAN.

### 3.3.3 Light quark dataset

Similarly to the gluon dataset, the EPiC-GAN and the MP-GAN both agree well with the light quark test dataset in the nine distributions presented in Fig. 4. For the particle-level distributions, both models replicate the transverse momentum and the pseudorapidity equally well, however the azimuthal angle spectrum is better represented by the EPiC-GAN. Investigating in detail the ordered leading particle transverse momentum spectra, we observe that the leading 1$^{\text{st}}$ and 5$^{\text{th}}$ $p_{\text{T}}^{\text{rel}}$ spectra are equally well reproduced by both models, while the low $p_{\text{T}}^{\text{rel}}$ tail of the 20$^{\text{th}}$ particle is better represented by the EPiC-GAN. The lower tail of the particle multiplicity spectrum appears to be slightly over-sampled by the MP-GAN. The relative jet mass distribution is well represented by both models, although both models appear to slightly over-sample the high mass tail. Both models perform equally well in the relative jet $p_{\text{T,jet}}^{\text{rel}}$ spectrum.

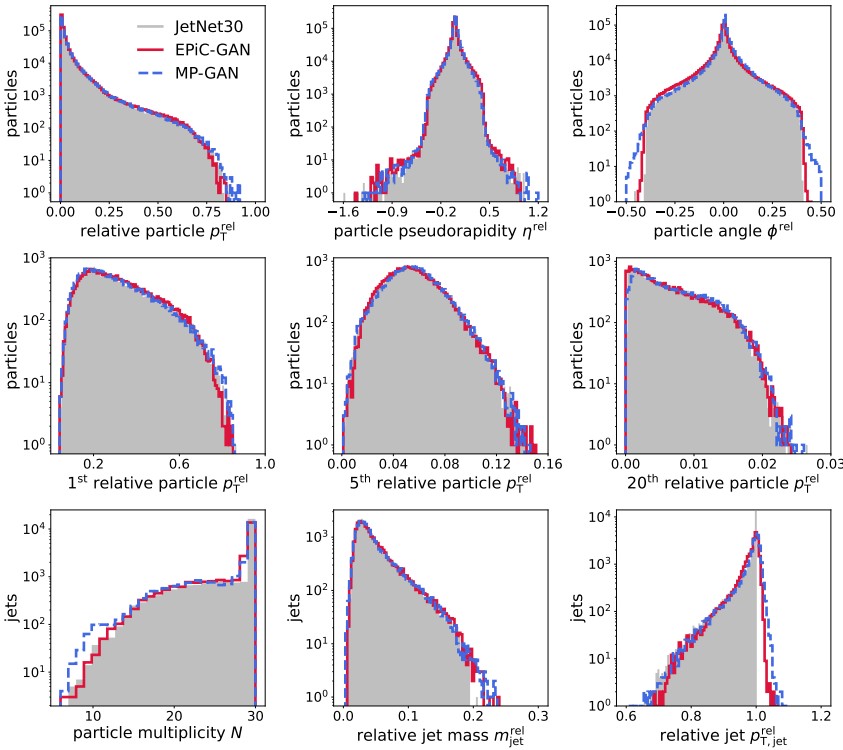

Figure 4: Same as Fig. 3 but for the JetNet30 light quark dataset.

### 3.3.4 Top dataset

Comparing the MP-GAN and the EPiC-GAN on the top quark dataset in Fig. 5 paints a similar picture as above: overall both models reproduce the nine distributions very well. Observing in detail the particle-level spectra, the $p_T^{rel}$ and $\phi^{rel}$ distributions are equally well represented by both GANs, while both struggle with a few outliers in the $\eta^{rel}$ spectrum. The 1$^{st}$ and 5$^{th}$ $p_T^{rel}$ spectrum are equally well represented by both models, while the 20$^{th}$ $p_T^{rel}$ distribution is slightly overestimated by the MP-GAN. For the jet-level observables, both the particle multiplicity and even the bimodal relative jet mass are captured well by both models. The challenging relative jet $p_{T,jet}^{rel}$ distribution is more closely reproduced by the EPiC-GAN than by the MP-GAN.

Overall we conclude that both MP-GAN and EPiC-GAN reproduce the JetNet30 datasets very well. When comparing the model scores to the truth scores in Table 2, though, neither model is quite reaching the fidelity of the dataset itself. In practice, GAN samples need to be verified to have a sufficient fidelity for a given physics analysis task. The metrics for judging the fidelity depend on the simulation task, and there are approaches that could increase the fidelity such as a learned re-weighting of generated samples [47]. Alternatively, important physics observables could be added to the loss function, i.e. in Ref. [20] the jet mass was added as a conditioning. However, even if certain observables are individually added to the loss function, their correlations might still be mismodeled. Therefore, using an unconditional model like the EPiC-GAN might be advantageous as certain correlations can be learned and directly encoded into the global attribute vector of the model, see Sec. 3.6.

## 3.4 JetNet150 results

Having observed competitive results with the EPiC-GAN on the JetNet30 datasets, we now show results for the more challenging JetNet150 dataset with up to 150 particles. We do not have a comparison with another generative model, since to our knowledge we are the first to

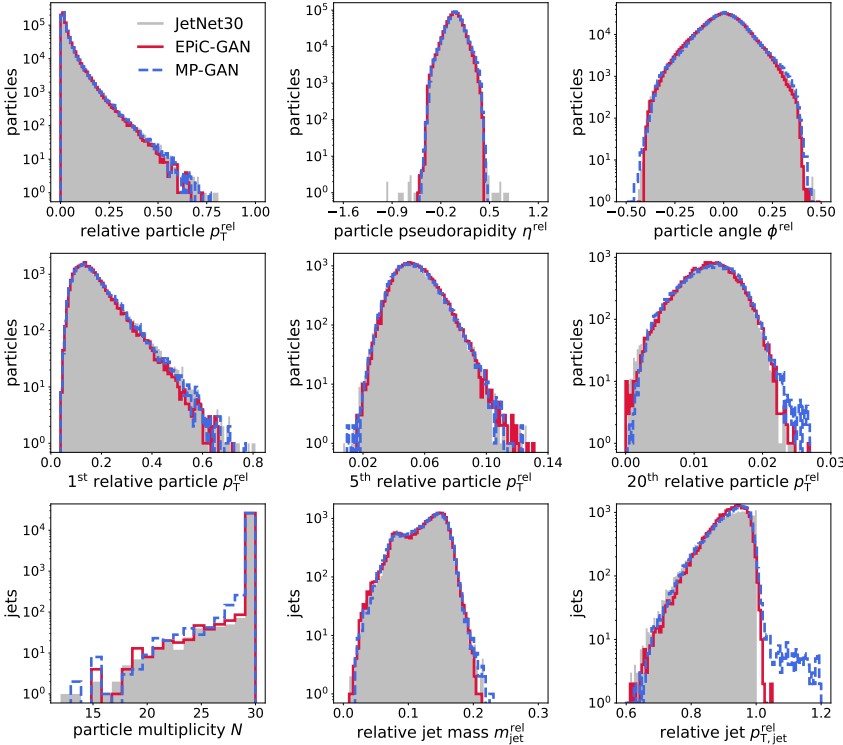

Figure 5: Same as Fig. 3 but for the JetNet30 top quark dataset.

show a well performing and fast generating model on a jet dataset with such large particle multiplicity.

The model architecture and training procedure is the same as for the JetNet30 datasets from Sec. 3.3. In the following, we comparing the EPiC-GAN results for the JetNet150 gluon, light quark and top datasets to the test dataset using the Wasserstein-1 distance metrics. We then show the previously discussed nine particle- and jet-level distributions for the JetNet150 top dataset, which is the most challenging of the three datasets.

In Table 3, we compare EPiC-GAN generated events to the JetNet150 truth with the three Wasserstein-1 distances introduced in Sec. 3.3. As of writing this publication, the FPND evaluation score was not available for the JetNet150 dataset. For both the gluon and the light quark dataset, the EPiC-GAN yields comparable results to the truth in the $W_1^M$ and $W_1^{EFP}$ scores. For the top dataset, the EPiC-GAN performs a bit worse in these observables. For all three datasets,

Table 3: Evaluation scores for the JetNet150 dataset. The truth values are a comparison between the test and training set. Lower is better for all scores.

| Jet class | Model | $W_1^M$ ($\times 10^{-3}$) | $W_1^P$ ($\times 10^{-3}$) | $W_1^{EFP}$ ($\times 10^{-5}$) |
|---|---|---|---|---|
| Gluon | Truth | $0.3 \pm 0.1$ | $0.3 \pm 0.1$ | $0.7 \pm 0.3$ |
| | EPiC-GAN | $0.4 \pm 0.1$ | $3.2 \pm 0.2$ | $1.1 \pm 0.7$ |
| Light quark | Truth | $0.3 \pm 0.1$ | $0.3 \pm 0.2$ | $0.6 \pm 0.5$ |
| | EPiC-GAN | $0.4 \pm 0.1$ | $3.9 \pm 0.3$ | $0.7 \pm 0.4$ |
| Top | Truth | $0.3 \pm 0.1$ | $0.2 \pm 0.1$ | $1.3 \pm 0.8$ |
| | EPiC-GAN | $0.6 \pm 0.1$ | $3.7 \pm 0.3$ | $2.8 \pm 0.7$ |

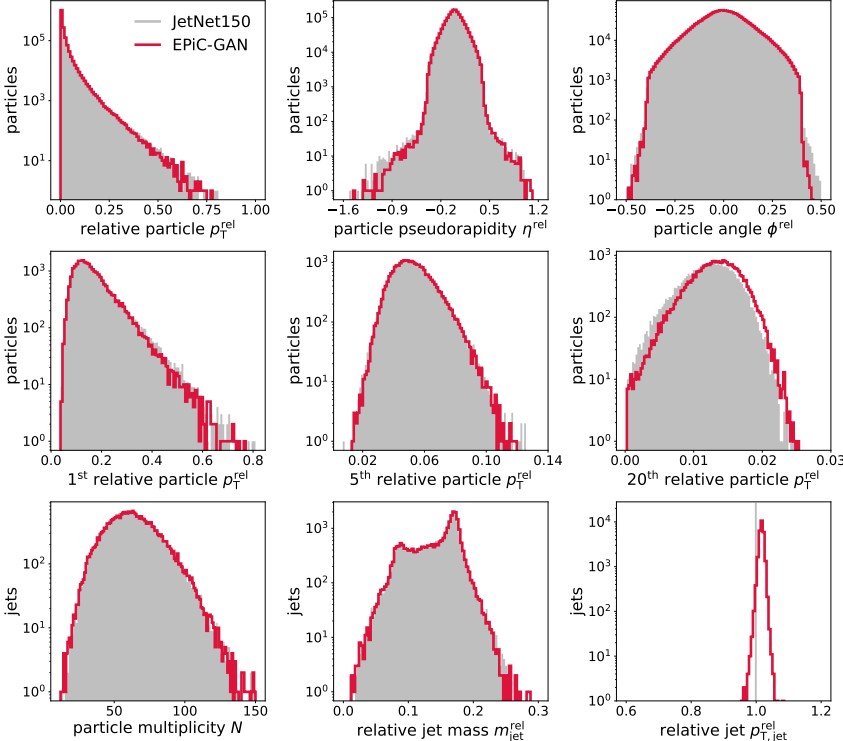

Figure 6: JetNet150 top dataset. Comparison between the test set and EPiC-GAN generated sets for various particle- and jet-level observables.

the EPiC-GAN performs worse than the truth for the particle feature score $W_1^P$.

It is understandable that the performance for the jet mass and the EFPs are quite good, since the trainings were chosen based on high jet mass fidelity and the EFPs are strongly correlated to the mass. Furthermore, the bimodal structure of the top mass distribution makes this dataset more challenging to generate than the other two. An improved training procedure that takes into account particle-level features when choosing an epoch, or an improved generative model, might yield even better results.

As for the JetNet30 dataset, the EPiC-GAN model reproduces the JetNet150 training data very well, though not perfectly. As mentioned in Sec. 3.5 below, the generation of 150 particles compared to 30 particles is only slower by a linear factor with the EPiC-GAN. Therefore, we are the first to provide results on this high cardinality dataset and are overall satisfied with the fidelity and generation speed of the EPIC-GAN.

To illustrate the fidelity of the EPiC-GAN on the JetNet150 top dataset, we provide in Fig. 6 a comparison of the nine physical distributions introduced in Sec. 3.3 between the test dataset and the same number of EPiC-GAN generated events. All the particle-level distributions are reproduced very well by the EPiC-GAN. The only exceptions are that the EPiC-GAN appears to slightly underestimate the $\phi^{\text{rel}}$ spectrum and to slightly overestimates the $20^{\text{th}}$ $p_T^{\text{rel}}$ distribution. The KDE-based particle multiplicity sampling agrees well with the base distributions, as does the bimodal jet mass spectrum that is particularly challenging to reproduce.

Due to the pre-processing outlined in Sec. 3.1, the relative jet $p_{T,\text{jet}}^{\text{rel}}$ spectrum resembles a single peak around $p_{T,\text{jet}}^{\text{rel}} = 1$, which is particularly difficult for a generative model such at the EPiC-GAN and therefore this spectrum is not modeled perfectly. This could be largely resolved, though, by repeating the calibration to $p_{T,\text{jet}}^{\text{rel}} = 1$ with the EPiC-GAN generated events — to illustrate this issue here, we have not performed this calibration. Overall, all top datasets' particle- and jet-level distributions are well represented by the EPiC-GAN model and the same

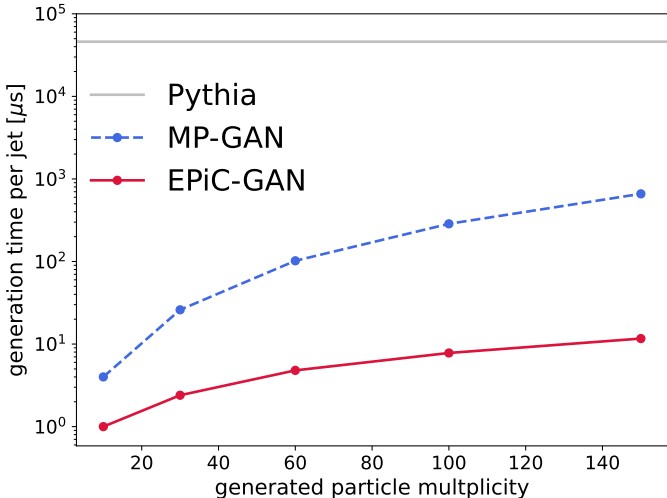

Figure 7: Generation time per jet as a function of generated particle multiplicity. For both GANs, a total of 500k jets were generated. The batch size was optimized for optimal generation speed, and generation was performed on a system with Intel Gold-5218 (64 CPUs @ 2.3 GHz), 384GB RAM, and NVIDIA A100-40GB. The Pythia baseline value is taken from Ref. [24].

is true when looking in detail at the same observables for the JetNet150 gluon and light quark datasets.

## 3.5 Timing

While the MP-GAN and EPiC-GAN achieve comparable generative fidelity, the advantage of the EPiC-GAN is faster generation. Relying on fully-connected message passing, the MP-GAN scales quadratic with the particle multiplicity $\approx \mathcal{O}(N^2)$. The EPiC-GAN does not utilize pairwise edge information, hence it scales linearly $\approx \mathcal{O}(N)$.

A comparison of the generation time per jet for different particle multiplicities between the MP-GAN and the EPiC-GAN on the same system is shown in Fig. 7. We studied the scaling behavior of the MP-GAN with its implementation from Ref. [24] and scanned the generation speed by adapting the particle multiplicity hyperparameter without training the model. The linear scaling behavior of the EPiC-GAN allows us to perform a jet generation study with the JetNet150 dataset, for which training the MP-GAN is too expensive. Further, the linear scaling of the EPiC-GAN also allows the model to be applicable to physics simulation tasks that are traditionally much more computationally expensive than the lightweight Pythia event generation, such as calorimeter shower simulation with Geant4 [48].

For 30 particles, the EPiC-GAN generates a jet 13x faster ($2\,\mu s$ vs. $26\,\mu s$), while for 150 particles 55x faster ($12\,\mu s$ vs. $660\,\mu s$). In comparison to the original Pythia generation time ($46\,ms$) [24], both GANs are significantly faster. When training the EPiC-GAN, one epoch takes on the same system about $65\,s$ for the JetNet30 top dataset (988 weight updates) and $85\,s$ for the JetNet150 top dataset (1058 weight updates).

## 3.6 Interpretability

Particle jets are defined by a number of per-jet (i.e. global) physical observables such as mass, transverse momentum, and particle multiplicity. This was a key motivation for the application of the EPiC layers, which utilize a global attribute vector $\boldsymbol{g}$ together with the set of points $P$. The EPiC-GAN should be able to use this vector $\boldsymbol{g}$ to encode physically meaningful jet

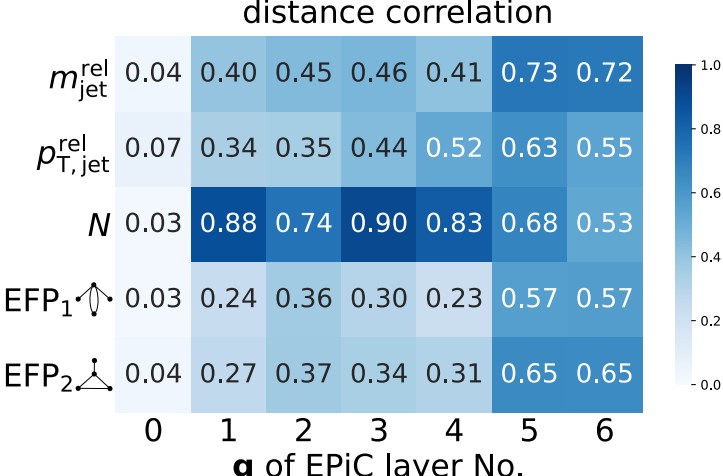

Figure 8: Distance correlation between a number of physical jet observables and the global attribute vector $\boldsymbol{g}$ after each EPiC layer for the JetNet30 quark training. This correlations allow us to interpret which physical features are encoded by the model into its global latent space.

features. As the vector $\boldsymbol{g}$ is combined with the point vectors $\boldsymbol{p}_i$ in the EPiC-layer, the points are transformed such that they make up a realistic particle jet.

To test whether the EPiC-GAN is indeed encoding physically meaningful jet features, we show in Fig. 8 the distance correlation [49] between a number of physical jet observables and the global attribute vector $\boldsymbol{g}$ for the JetNet30 light quark dataset. The correlations are calculated from 5,000 generated events for which the global attributes after each EPiC-layer were compared to the calculated observables from the corresponding events. The physical observables shown are the relative jet mass $m_{\text{jet}}^{\text{rel}}$, the relative jet transverse momentum $p_{\text{T,jet}}^{\text{rel}}$, the particle multiplicity $N$, and two EFPs (out of the five EFPs used to calculate $W_1^{\text{EFP}}$ introduced in Sec. 3.3).

One can see that after each EPiC layer, there is a significantly higher distance correlation of $\boldsymbol{g}$ with all physical observables than before the first EPiC layer (corresponding to the sampled noise). Therefore, it appears that these physical observables are indeed partly encoded into the global attribute space. Observing these distance correlations allows us to monitor and interpret which physical features are well learned by the EPiC-GAN.

## 4 Conclusions

Many tasks in physics and other sciences involve data that can be best represented as point clouds or sets. This is especially the case when the data results from measurements taken in a heterogeneous geometry with a variable cardinality of measured features and data points. One example of such data are particle jets with either four-vectors or hadronic coordinates as features and a variable particle multiplicity. Previous state-of-the-art generative models for point cloud data in HEP considered graph- and transformer-based networks. These architectures are computationally expensive when scaled to large jet sizes, however, which limits their application to simulating complex collider events in high resolution calorimeters.

In this paper, we introduced a simple, yet high fidelity set-based alternative to graph-based generative models. Our GAN framework utilizes Equivariant Point Cloud (EPiC) layers in the generator and discriminator. The resulting EPiC-GAN is a permutation equivariant generative

model that allows for variable particle multiplicity, and its computation cost scales linearly with the particle multiplicity per generated events.

As a case study, we compared EPiC-GAN to the state-of-the-art network on the JetNet30 benchmark dataset: MP-GAN. We observed comparable generative fidelity between the two models, yet significantly faster generation time for EPiC-GAN. Furthermore, the EPiC-GAN scales well to large point clouds sizes, as demonstrated by the JetNet150 results, whereas the generation time of MP-GAN scales quadratically with the number of points. Considering that for HEP analyses, millions of simulated events over various channels are often necessary, a fast and computationally cheap generation is of particular importance. Depending on the application, i.e. for proof-of-principle studies or large scale parameter scans, it may even be beneficial to opt for a model with slightly worse fidelity if it comes at a significantly lower cost.

One might have expected that a fully-connected GAN based on message-passing would generalize better to complicated point clouds than a simpler set-based approach. For the jet datasets presented here, however, this complexity seems not necessary or is not utilized. This might be due to the moderately complex topology of jets investigated here: the gluon and light quark jets have one-prong structure and the top jets has a three-prong structure due to its decay via a *W* boson into three light quarks. We speculate that due to their high degree of randomness and relatively small number of relevant features, point clouds in particle physics can generally benefit from the simple protocol proposed in this work, where inter-point communication is facilitated by a global feature vector.

Alternatively, it may be that the GAN training procedure is limiting the possible performance of both the graph- and set-based generative models. An increased stability of the GAN training or another generative framework, such as score-based models [50], might yield even better results. As the HEP community explores different generative training procedures, we encourage systematic studies of the relative importance of point, edge, and global features. Even outside of the GAN framework, we hope that EPIC layers will yield competitive generative fidelity with efficient scaling to large point clouds.

# Acknowledgements

We would like to thank the Maxwell computing center at DESY for the smooth operation and technical support.

**Funding information**    E. B. is funded by a scholarship of the Friedrich Naumann Foundation for Freedom and by the German Federal Ministry of Science and Research (BMBF) via *Verbundprojekts 05H2018 - R&D COMPUTING (Pilotmaßnahme ErUM-Data) Innovative Digitale Technologien für die Erforschung von Universum und Materie*. J.T. was supported by the U.S. Department of Energy (DOE) Office of High Energy Physics under Grant Contract No. DE-SC0012567 and by the National Science Foundation under Cooperative Agreement No. PHY-2019786 (The NSF AI Institute for Artificial Intelligence and Fundamental Interactions, http://iaifi.org/). E.B and G.K. acknowledge support by the Deutsche Forschungsgemeinschaft (DFG, German Research Foundation) under Germany's Excellence Strategy - EXC 2121 "Quantum Universe" - 390833306. This work was supported within the framework of the PIER Hamburg-MIT Germany program funded by the Ministry of Science, Research and Equalities and Districts of the Free and Hanseatic City of Hamburg.

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
