# Peer review of "EPiC-GAN: Equivariant Point Cloud Generation for Particle Jets"

_SciPost Physics, doi:SciPost Phys. 15, 130 (2023)_

## Round 2 · Referee Report · Anonymous (Referee 1) · 2023-4-4

Strengths

1 - Introduction of a novel general adversarial network architecture, which
delivers significantly faster computation times than previous approaches.

2 - Validation with a relevant physics case study.

3 - Relevant code is made available fully public on GitHub.

Weaknesses

1 - Little evaluation of literature critical of the general approach.

2 - No significant interpretation is given to the results obtained for the data
sets in the second part of the paper.

Report

The authors introduce a novel generative adversarial network architecture
that is demonstrated to offer significant advantages in computation time over
previous approaches. This method of data generation is validated in a relevant
physics setup of generating particle jets with high multiplicities based on
available training data sets.

The approach is well motivated and documented clearly and sufficient in the
manuscript. The improved algorithmic complexity and resulting faster computation
times are promising and as such deserve publication in my opinion. However, I
would like the authors to comment more critically on the general applicability of the approach and their achieved results. Hence, I recommend to ask the authors to revise their manuscript on some minor points before publication.

Requested changes

1 - For example SciPost Phys. 12 (2022) 3, 104 raised questions toward the achievable statistical power of samples generated from adversarial networks compared to the corresponding training samples. While I understand that addressing such questions is not a goal of the study, I feel the authors should comment in how far these limitations apply to their approach, or how they would be avoided.

2 - More pragmatically, how would one practically deal with the still existing
deviations between reference data set and generated data visible in some of the observables (in Figs. 3-6)? Can the authors give a concrete recommendation with what observables one can trust their generative network without the need to check explicitly, or ideally provide a prescription to assign an uncertainty that would guarantee agreement with any test data?

3 - I generally would appreciate a more detailed analysis of the differences
observed between the different data sets and what can be learned from
them. Currently the text simply notes these deviations for the individual
sets. However those can of course be readily be read off from the figures, while
little to no actual interpretation is given. For example, relative jet
transverse momentum appears to be reproduce well for both light and top quark jets, but significantly deviates for gluons. Is this understood and can anything be learned from this?

  • validity: -
  • significance: -
  • originality: -
  • clarity: -
  • formatting: -
  • grammar: -

Author:  Erik Buhmann  on 2023-07-17  [id 3815]

(in reply to Report 1 on 2023-04-04)
Category:
remark
answer to question
correction

We would like to thank the reviewer for their detailed comments and for giving us the opportunity to further improve our manuscript.

In response to the minor points raised:

1 - We agree that the successful application of a GANs in HEP are rather problem dependent. As SciPost Phys. 12 (2022) 3, 104 states, a GAN can be used for certain HEP applications, but possible uncertainties introduced need to be carefully examined. We have added a mention of this issue to the 3rd paragraph of the introduction, namely: “Here, we introduce a novel kind of GAN architecture which could be applied to various applications. Of course, when applied to a specific physics analysis, the uncertainties derived from a generative model need to be critically examined depending on the use-case.”

2 - Assigning error bars to the GAN outputs would be very interesting, but such work is beyond the scope of this paper due to computational and statistical constraints. Further, our submission focuses on model development with a case study based on the JetNet dataset, which is a good benchmark for the developments of high fidelity models. We added to the end of Section 3.3.4 that, “In practice, GAN samples need to be verified to have a sufficient fidelity for a given physics analysis task. The metrics for judging the fidelity depend on the simulation task, and there are approaches that could increase the fidelity such as a learned re-weighting of generated samples. Alternatively, important physics observables could be added to the loss function, i.e. in Ref. [20] the jet mass is added as a conditioning. However, even if certain observables are individually added to the loss function, their correlations might still be mismodeled. Therefore, using an unconditional model like the EPiC-GAN might be advantageous as certain correlations can be learned and directly encoded into the global attribute vector of the model.”

3 - As mentioned in Section 3.2, we observed a few variations between different training runs due to the unstable nature of adversarial training. We did not observe consistent mismodelling of certain variables, rather slight fluctuations with certain observables being slightly better modeled than others. Due to the model choice based on the jet mass, other observables might not always be optimal. We decided to leave better model choices to further work to compare directly our GAN to the MPGAN previously introduced. Due to the statistical fluctuations between model choices and our model being developed as a general purpose point cloud generative model, we don’t expect to learn much about the dataset from the way certain observables are modeled.

---

## Round 2 · Referee Report · Tilman Plehn (Referee 2) · 2023-5-17

Report

The paper is, technically innovative and very interesting, shows impressive results on jet generation, and should definitely be published in SciPost Physics. I have only a few comments on the presentation and the context.

  • on p.2, the authors emphasize a `strong inductive bias' as a requirement for generative networks. I know the papers they are referring to, but I disagree with this statement, because it sounds like generative networks have to optimized to predict phase space ensities. But that has never been shown, and instead we know that a wide range of models can be used for phase space generation.

  • I strongly disagree with your statement that the most advanced models are applied to calorimeter showers. For instance, the normalizing flows we are using for event generation and unfolding are more powerful than the corresponding flows for the calorimeter showers. Plus, when it comes to control, precision, and uncertainty estimation the events applications are technically far ahead. Please change that, every event generation paper cites calorimeter showers out of fairness and professionality.

  • you are not citing CaloMan, even though I think this is a very cool paper? Please check that reference list for completeness as well, some classic GANs on showers etc seem to be missing as well.

  • most importantly, I am missing a paragraph on the physics application. You are generating showers, which I would, naively, not consider a bottleneck in speed or precision of LHC simulations. What is your take on that? If you are using this application just as a toy example, please state this clearly to avoid confusion. Your names will otherweise lead people to assume that this is an application needed for HL-LHC, and Fig.7 suggests that we have a problem with Pythia's speed.

  • any chance you would want to move some of the architecture aspects or hyperparameters from the text to a table? To make it easier to find the relevant information?

  • do you have any intuition why the LSGAN is already stable and outperforms the cross entropy loss? And you do not need any additional ways to stabilize the GAN training?

  • as you, I am critical of the stopping criterion linked to a hand-picked high-level observables. One way to justify this would be to save networks during training and show a set of distributions for each network. What happens then?

  • I am also sceptical concerning the different ways to compare the different generative datasets, as introduced in [17]. I am fine using their set, but please also train a standard classifier, see if the ROC curves look as expected, maybe show the ROC curves for a set of trainings to estimate the stability, and quote the AUC. It's the one metric where your readers have an intuition.

  • for the evaluation scores, are you using 10 generated datasets from the same training, or from different trainings? I would think the second is what we want.

  • please add an error bar from your 10 independent trainings (I hope) to the distributions of Fig.3 etc. And maybe secondary panels with bin-wise ratios or something line that.

  • in your conclusion, you suggest that fast and less precise simulatons are fine for the LHC. Of course you entitled to say that, but I would like to explicitly state that I very strongly disagree, as would the precision-QCD community and Gavin Salam's group working on precision showers.

Aside from that, I think the paper is very cool and should be published, after implementing a few modifications and clarifications.

  • validity: top
  • significance: high
  • originality: high
  • clarity: top
  • formatting: perfect
  • grammar: perfect

Author:  Erik Buhmann  on 2023-07-17  [id 3816]

(in reply to Report 2 by Tilman Plehn on 2023-05-17)
Category:
remark
answer to question
correction

We would like to thank the reviewer for their praise and their detailed comments.

In response to the points raised: 1 - Regarding the “strong inductive bias”: Indeed, many model architectures can be used as generative models and all of them carry some kind of inductive bias which helps the model learn something about the underlying data distribution. We did not want to imply anything out of the ordinary about the EPiC-GAN, so we have softened our statement, now stating: “For these applications, the generative models need to learn the underlying data distributions with help from the inductive bias of the model architecture.”

2 - Regarding “most advanced models”: Indeed, the task between generating calorimeter showers and generating phase space events are too different to make definitive statements about which are more advanced. We’ve softened the language and have highlighted that advanced generative modeling efforts are undertaken to generate high-dimensional HEP data. The manuscript now states: “Some very advanced generative modeling efforts for high-dimensional data in high energy physics (HEP) are currently undertaken for calorimeter shower simulations.”

3 - Citations: We added the following citations to the introduction: – S. Vallecorsa, F. Carminati and G. Khattak, 3D convolutional GAN for fast simulation, EPJ Web Conf. 214, 02010 (2019) – M. Erdmann, J. Glombitza and T. Quast, Precise Simulation of Electromagnetic Calorimeter Showers Using a Wasserstein Generative Adversarial Network, Computing and Software for Big Science 3(1) (2019) – G. R. Khattak, S. Vallecorsa, F. Carminati and G. M. Khan, Fast Simulation of a High Granularity Calorimeter by Generative Adversarial Networks (2021), arXiv 2109.07388 – ATLAS Collaboration, Deep generative models for fast photon shower simulation in ATLAS (2022), arXiv 2210.06204 – E. Buhmann, S. Diefenbacher, E. Eren, F. Gaede, G. Kasieczka, A. Korol and K. Krüger, Decoding Photons: Physics in the Latent Space of a BIB-AE Generative Network, EPJ Web of Conferences 251, 03003 (2021) – J. C. Cresswell, B. L. Ross, G. Loaiza-Ganem, H. Reyes-Gonzalez, M. Letizia and A. L. Caterini, CaloMan: Fast generation of calorimeter showers with density estimation on learned manifolds (2022), arXiv 2211.15380

4 - Application: The application of the EPiC-GAN here is to the JetNet benchmark dataset, which is a toy dataset for the comparison of point cloud generative models in particle physics. We have added additional clarifications on this point to the introduction. Further, we have added that one could envision fast point cloud generative models to be applied to background estimations for anomaly detection or to nuisance parameter tuning. It reads now in the introduction: “JetNet is a specifically designed toy dataset for the generation of hadronized jets used to compare point cloud generative models.” and “We further envision that our model could be applied to generate in-situ background events for anomaly detection methods such as CATHODE [34]. Additionally, the development of fast and lightweight generative models can support analysis by making Monte Carlo tuning or nuisance parameter variation computationally efficient.“ Additionally, we added to section 3.5: “Further, the linear scaling of the EPiC-GAN also allows the model to be applicable to physics simulation tasks that are traditionally much more computationally expensive than the lightweight Pythia event generation, such as calorimeter shower simulation with Geant4.”

5 - Hyperparameter table: Good suggestion, this streamlined Section 3.2.

6 - Training stability: In addition to the LSGAN, we apply weight normalization to all hidden layers (added to Section 2.2). It now reads: “We compared this approach to the regular GAN objective using the binary cross-entropy loss and found slightly improved training stability with the LSGAN objective as it avoids vanishing gradients. Additionally, we apply weight normalization to all hidden layers of both the generator and discriminator to improve the GAN stability.”

7 - Stopping criterion based on jet mass: We checked that multiple observables are correlated with a good mass modeling (as mentioned in the manuscript, EFPs are correlated with the mass and therefore a good mass yields high fidelity in other observables).

8 - Classifier ROC curve: A classifier training would indeed be an interesting additional evaluation metric. Unfortunately we don’t have enough training statistics to facilitate the training of such a classifier. We set aside 15% (or 25,000) test events for model evaluation, but for a classifier training we’d need significantly more training data (that is not used in the generator training).

9/10 - Evaluation scores: We have one final model trained for each jet class. We use this model to generate 10x the test statistics, i.e. 250k events. The scores are now calculated as the mean of the Wasserstein distance of an observables calculated from the 25k test set and one generated set of 25k events. Therefore, the mean over 10 Wasserstein distances is reported. This is consistent with the training procedure of Ref. [17], where the mean over 5 Wasserstein distances of each 10k events (sampled from 50k test and generated events via bootstrapping) is reported. Since we used a smaller test set, we increased the amount of generated statistics. For the MPGAN scores in our manuscript, we re-calculated all scores with the increased statistics using the trained weights published alongside Ref. [17].

11 - Required fidelity depending on the application: We do not state or mean to imply that no precision simulations are or will be needed. We say that for each application – e.g. each data analysis stream – a different level of fidelity is required. We still think this is a true statement and hope neither you nor Gavin would believe that one could advocate for low-precision simulations for measurements. For clarification, the sentence in the conclusion now reads: “Depending on the application, i.e. for proof-of-principle studies or large scale parameter scans, it may even be beneficial to opt for a model with slightly worse fidelity if it comes at a significantly lower cost.”

For completion, these additional changes have been made to the manuscript: – We recalculated the MPGAN metrics, now with the same post processing applied as to the EPiC GAN (centering of jets). The values changed just slightly within the margin of error for Gluon W1EFP from 0.6 +- 0.4 to 0.6 +- 0.3; for Top W1M from 0.4 +- 0.1 to 0.5+- 0.1; and for Top W1EFP from 0.9 +- 0.3 to 1.0 +- 0.7. – Figure 2: In the discriminator diagram the aggregation was supposed to be in front of the MLP phi_p^in, not behind. – Added to Sec. 3.2.: “In principle, the discriminator could also be implemented with any other permutation-equivariant graph- or set-based architecture. However, replacing the EPiC discriminator with a simple Particle Flow Network -- equivalent to L_discriminator=0 EPiC layers -- led to worse results.”

---

## Round 3 · Referee Report · Anonymous (Referee 1) · 2023-7-18

Report

I am satisfied with the authors replies to my comments.

---

## Round 3 · Referee Report · Tilman Plehn (Referee 2) · 2023-7-20

Report

Thank you for considering all my comments. Concerning the references, I am unhappy, because declaring detector-generative models and event-generative models too different to cite each other is not good for the scientific exchange and not good for the careers of theory students. For instance, the Rutgers and Heidelberg groups just wrote a paper combining calorimeter flows and event flows, illustrating the fruitful combination of the two task. In addition, are we not pretending that this exchange is really important for our grant applications? But this is clearly a choice of the authors, fine by me.

---

## Round 3 · Author Response

We thank our colleagues for their thoughtful comments on the manuscript and the opportunity to resubmit. We have already provided direct responses to the reviewer comments. A detailed list of changes is provided as well.

---

## Round 3 · List of Changes

1- Softened our language in the Introduction: “For these applications, the generative models need to learn the underlying data distributions with help from the inductive bias of the model architecture.”

2- Softened our language in the Introduction: “Some very advanced generative modeling efforts for high-dimensional data in high energy physics (HEP) are currently undertaken for calorimeter shower simulations.”

3- Added to the third paragraph of the Introduction: ““Here, we introduce a novel kind of GAN architecture which could be applied to various applications. Of course, when applied to a specific physics analysis, the uncertainties derived from a generative model need to be critically examined depending on the use-case.”

4- We expanded on the application in the Introduction: “JetNet is a specifically designed toy dataset for the generation of hadronized jets used to compare point cloud generative models.” and “We further envision that our model could be applied to generate in-situ background events for anomaly detection methods such as CATHODE [34]. Additionally, the development of fast and lightweight generative models can support analysis by making Monte Carlo tuning or nuisance parameter variation computationally efficient.“

5- Citations: We added the following citations to the Introduction: – S. Vallecorsa, F. Carminati and G. Khattak, 3D convolutional GAN for fast simulation, EPJ Web Conf. 214, 02010 (2019) – M. Erdmann, J. Glombitza and T. Quast, Precise Simulation of Electromagnetic Calorimeter Showers Using a Wasserstein Generative Adversarial Network, Computing and Software for Big Science 3(1) (2019) – G. R. Khattak, S. Vallecorsa, F. Carminati and G. M. Khan, Fast Simulation of a High Granularity Calorimeter by Generative Adversarial Networks (2021), arXiv 2109.07388 – ATLAS Collaboration, Deep generative models for fast photon shower simulation in ATLAS (2022), arXiv 2210.06204 – E. Buhmann, S. Diefenbacher, E. Eren, F. Gaede, G. Kasieczka, A. Korol and K. Krüger, Decoding Photons: Physics in the Latent Space of a BIB-AE Generative Network, EPJ Web of Conferences 251, 03003 (2021) – J. C. Cresswell, B. L. Ross, G. Loaiza-Ganem, H. Reyes-Gonzalez, M. Letizia and A. L. Caterini, CaloMan: Fast generation of calorimeter showers with density estimation on learned manifolds (2022), arXiv 2211.15380

6- Figure 2: In the discriminator diagram the aggregation was supposed to be in front of the MLP phi_p^in, not behind.

7- Expanded on the end of Sec. 2.2: “We compared this approach to the regular GAN objective using the binary cross-entropy loss and found slightly improved training stability with the LSGAN objective as it avoids vanishing gradients. Additionally, we apply weight normalization to all hidden layers of both the generator and discriminator to improve the GAN stability.”

8- We added and streamlined Sec. 3.2. with a Table of the hyperparameters.

9- Added to Sec. 3.2.: “In principle, the discriminator could also be implemented with any other permutation-equivariant graph- or set-based architecture. However, replacing the EPiC discriminator with a simple Particle Flow Network -- equivalent to L_discriminator=0 EPiC layers -- led to worse results.”

10-Table 2: We recalculated the MPGAN metrics, now with the same post processing applied as to the EPiC GAN (centering of jets). The values changed just slightly within the margin of error for Gluon W1EFP from 0.6 +- 0.4 to 0.6 +- 0.3; for Top W1M from 0.4 +- 0.1 to 0.5+- 0.1; and for Top W1EFP from 0.9 +- 0.3 to 1.0 +- 0.7.

11- Added to the end of Sec. 3.3.4: “In practice, GAN samples need to be verified to have a sufficient fidelity for a given physics analysis task. The metrics for judging the fidelity depend on the simulation task, and there are approaches that could increase the fidelity such as a learned re-weighting of generated samples. Alternatively, important physics observables could be added to the loss function, i.e. in Ref. [20] the jet mass is added as a conditioning. However, even if certain observables are individually added to the loss function, their correlations might still be mismodeled. Therefore, using an unconditional model like the EPiC-GAN might be advantageous as certain correlations can be learned and directly encoded into the global attribute vector of the model.”

12- We expanded on the application in Sec. 3.5: “Further, the linear scaling of the EPiC-GAN also allows the model to be applicable to physics simulation tasks that are traditionally much more computationally expensive than the lightweight Pythia event generation, such as calorimeter shower simulation with Geant4.”

13- Clarified in the Conclusion: “Depending on the application, i.e. for proof-of-principle studies or large scale parameter scans, it may even be beneficial to opt for a model with slightly worse fidelity if it comes at a significantly lower cost.”

---

## Editorial Decision

published